# Branched Setae or Attached Macroalgae: A Case Study of an Exceptionally Preserved Brachiopod from the Early Cambrian Chengjiang Lagerstätte

**DOI:** 10.3390/biology14091287

**Published:** 2025-09-18

**Authors:** Yue Liang, Timothy P. Topper, Baopeng Song, Caibin Zhang, Oluwatoosin B. A. Agbaje, Lars E. Holmer, Zhifei Zhang

**Affiliations:** 1State Key Laboratory of Continental Evolution and Early Life, Shaanxi Key Laboratory of Early Life & Environments, Department of Geology, Northwest University, Xi’an 710069, China; yue.liang@nwu.edu.cn (Y.L.); timothy.topper@nwu.edu.cn (T.P.T.); baopeng_nwu@stumail.nwu.edu.cn (B.S.); caibin-nwu@foxmail.com (C.Z.); lars.holmer@pal.uu.se (L.E.H.); 2Department of Palaeobiology, Swedish Museum of Natural History, Box 50007, SE-104 05 Stockholm, Sweden; 3Globe Institute, University of Copenhagen, Øster Voldgade 5-7, 1350 Copenhagen, Denmark; oluwatoosin.agbaje@sund.ku.dk; 4Department of Earth Sciences, Palaeobiology, Uppsala University, SE-752 36 Uppsala, Sweden

**Keywords:** Cambrian, brachiopod, setae, macroalgae

## Abstract

This study explores an exceptionally well-preserved fossil called *Xianshanella haikouensis*, a brachiopod species from the early Cambrian period, discovered in Chengjiang biota, South China. The fossil exhibits unusual, branching hair-like structures, which may represent either brachiopod setae or attached macroalgae. Using several advanced imaging techniques, researchers found that these structures are preserved as iron oxides, typical of Chengjiang fossil preservation. The discovery raises questions about whether these structures are branched setae of the brachiopod or evidence of an early symbiotic relationship with macroalgae. If confirmed as brachiopod setae, this finding would advance our understanding of the evolutionary links between brachiopods and annelids, shedding light on how early life forms diversified. Alternatively, if they are macroalgae, it would highlight the ecological complexity of marine ecosystems during the Cambrian period, showing how organisms interacted and relied on one another for survival. This research not only deepens our knowledge of early animal evolution but also underscores the importance of exceptional fossil preservation in uncovering the mysteries of life’s history.

## 1. Introduction

Exceptionally preserved fossils from the Cambrian Explosion have greatly improved our understanding of animal origin and evolution. Brachiopoda, firmly nested within the lophotrochozoan protostomes, is one of the most successful invertebrate phyla in terms of abundance and diversity during the Phanerozoic Eon [1,2,3,4,5]. The large majority of brachiopod genera are known almost exclusively from their mineralized shell valves, while our knowledge of brachiopod anatomy, notably soft tissues and organs, relies heavily on their extant relatives [6]. Fortunately, the discovery of exquisitely preserved soft tissues in exceptional Cambrian deposits, particularly the celebrated Chengjiang and Burgess Shale Konservat-Lagerstätten, has provided unprecedented insights into shell interior morphology and anatomy [3,7,8].

Shared morphological characters among lophotrochozoans are primarily restricted to embryonic and larval features [9,10]. Phylogenetic analyses identify a subsidiary trochozoan clade comprising annelids, brachiopods, and mollusks, in which chitinous setae (or chaetae) represent a putative ancestral character [11,12,13,14]. These chitinous projections act a range of sensory and locomotory functions and have been regarded as a key character in phylogenetic analyses [11,15,16,17,18,19,20,21]. The morphological similarities between annelid and brachiopod setae have been interpreted as homologous structures [20,21,22,23], which is supported by recent Hox gene research [14,24]. The recent discovery of chitinous setae bundles in the early Cambrian helcionelloid *Pelagiella* provides the first soft-tissue evidence supporting its interpretation as a stem-group gastropod [25]. Accordingly, the presence of chitinous setae in early Cambrian groups of lophotrochozoans (particularly brachiopods and annelids, as well as stem groups such as tommotiids and mickwitziids) provides critical evidence for their evolutionary origins, suggesting these structures may constitute a fundamental synapomorphy for lophotrochozoan evolution [13].

During the Cambrian period, brachiopods served dual ecological roles: they attached to other organisms while also functioning as substrates for epibionts, and this mutualistic behavior contributed significantly to increasing ecological complexity during the establishment of Earth’s first consumer-driven marine ecosystem [8,26]. For example, paterinid brachiopods with long and delicate setae attached to sponges in the Burgess Shale exhibit a mimicry system: the brachiopods (mimics) physically resemble their sponge hosts (models), which may have served to deceive predators or reduce detection [7,27,28]. More recently, early Silurian rhynchonelliform brachiopods with long and slender setae have been documented, exhibiting a statistically significant checkerboard distribution pattern correlated with setal length [29]. This spatial organization suggests setae played a crucial role in maintaining individual spacing and population structure on Paleozoic seafloors [29]. This fossil evidence uncovers a new mechanism for deep-time ecosystem structuring, showing notable ancient ecological impacts of subtle anatomical features.

Here, we describe novel, enigmatic branched fringes and delicate marginal setae in the lingulid brachiopod *Xianshanella haikouensis* Zhang and Han 2004 [30], from the early Cambrian Chengjiang Lagerstätte. Micro-CT reconstructions reveal these branched fringes distributed along the distal ends of marginal setae across the entire shell valve. These structures may represent either (1) branched setae, or (2) macroalgae attached to the brachiopod. While current evidence better supports the macroalgal attachment hypothesis, definitive resolution requires additional well-preserved specimens and advanced imaging techniques.

## 2. Materials and Methods

The specimen of *Xianshanella haikouensis* (specimen number: ELI 1781) was recovered from Haikou, Kunming, South China. The sample was obtained from a 25–30 m thick finely laminated mudstone with sporadic fossiliferous, thin (3–5 mm) siltstone, occasionally with intercalated 10–20 cm thick sandstone interbeds. Details of the localities and stratigraphy were given in Zhang et al. [31]. The specimen is deposited in the Early Life Institute and Department of Geology, Northwest University, Xi’an, Shaanxi Province, China.

The specimen was examined using a Zeiss Stemi 305 microscope and photographed with a Zeiss SmartZoom 5 stereomicrographic system equipped with a Canon camera to document overall morphology and surface details (Carl Zeiss Microscopy GmbH, Jena, Germany). Further analyses were conducted with an FEI Quanta 400 FEG-SEM (Thermo Fisher Scientific, Hillsboro, OR, USA) (20.0 kV, 60 Pa, Working Distance 8–10 mm) equipped with backscattered and secondary electron (SEM) detectors along with an energy-dispersive X-ray analyzer (EDS) to examine ultra-fine morphological features and elemental composition of branched fringes. Overall elemental characterization was performed using a Bruker M4 Tornado Micro X-ray fluorescent spectroscopy (Micro-XRF) (Bruker Corporation, Berlin, Germany) to determine elemental abundances and preservation mode. Micro-computed tomography (Micro-CT) analysis was carried out with an Xradia 520 Versa system (Carl Zeiss Microscopy GmbH, Jena, Germany), with 3D reconstructions and visualizations generated using Dragonfly 4.1 (ORS, Montreal, Quebec, Canada) to non-destructively investigate fossil details covered by matrix. All of the above experiments and analyses were conducted in the State Key Laboratory of Continental Evolution and Early Life, Northwest University.

## 3. Results

The shells of *Xianshanella haikouensis* are biconvex, equivalved, and rounded in outline. The valves reach a width of 8.3 mm and a length of 8.7 mm (Figure 1A), smaller than other individuals presented in Zhang et al. [31] with an average diameter of 15.2 mm. The specimen is characterized by exquisitely preserved branched fringes and marginal setae. Marginal setae are closely spaced and radiating outward from the shell margin, extending up to 5.8 mm. The branched fringes at the anterior end of marginal setae become elongated anteriorly, while those at the lateral part of the shell are curved and intersected with each other (Figure 1B–E). The branched fringes and their basal stems exhibit similar diameters, with a mean fringe diameter of 53.3 μm (Figure 1B–E). The widest branches, located in the anterior region, measure up to 60.5 μm (Figure 1B,C). While the marginal setae are comparatively slimmer with a mean diameter about 38.7 μm and the thinnest has a diameter of only 24.1 μm (Figure 1A,F). The branched fringes not only show subsequent dichotomous (second), but also third and fourth bifurcation (Figure 1B–E, Figure 2, Figure 3 and Figure 4). The first bifurcation occurs 1.7–3.3 mm above the shell margin (Figure 1B,C). Secondary branches consistently deviate at 20–45° angles from the primary spine axis (Figure 1B,C). Due to dorsoventral compression of the valves, setae from both shells are preserved in the same plane, making it challenging to distinguish between dorsal and ventral setal arrangements (Figure 1A,F).

Although the branched fringes are well-preserved in *X*. *haikouensis*, other key anatomical features, including the stout pedicle, spiral lophophore, and mantle canals, remain obscured from plan view. Fortunately, Micro-CT scanning successfully revealed both enhanced details of the branched fringes and previously obscured anatomical structures within the matrix, including the pedicle and lophophore (Figure 2). The reconstructions clearly show elongated, interwoven branched fringes along the right lateral shell region (Figure 2), a stout annulated pedicle (Figure 2B), and distinguishable spiral lophophore morphology (Figure 2A,C).

The specimen is preserved as a dark reddish imprint (Figure 1). Both Micro-XRF and EDS mapping analyses indicate that the shell valve, branched fringes and marginal setae are preserved by iron oxides (Figure 3A,B,D), contrasting to the matrix composed of aluminosilicate (Figure 3C). Detailed EDS point analyses show an overall similar chemical composition between the branched fringes and marginal setae (Figure 4A–L). These data reveal abundant magnesium (Mg), silicon (Si), and aluminum (Al) in both structures, with comparatively lower potassium (K) concentrations (Figure 4J–L). Meanwhile, phosphorus (P) was below detection limits in the marginal setae, carbon (C) was undetectable in either structure (Figure 4J–L). The branched fringes showed slightly elevated oxygen (O) levels relative to the marginal setae (Figure 4J–L).

## 4. Discussion

### 4.1. Branched Fringes as Elongate Setae and Its Phylogenetical Implications

The material of *Xianshanella haikouensis* exhibits Burgess Shale-type preservation of non-mineralized tissues in the fashion typical of the Chengjiang deposits [32,33,34,35,36,37,38]. Both branched fringes and marginal setae are preserved as iron oxides indicated by a high elemental abundance of iron compared with the matrix in Micro-XRF and EDS observations (Figure 3 and Figure 4). The shells of *X. haikouensis* are thought to have been originally organophosphatic, but the original shell material has been completely replaced by thin films of iron oxides, which possibly results from the relatively lower degree of mineralization of the shells. It is significant that Ca was not detected, and a likely interpretation is that the alteration of pyrite to iron oxides released small quantities of sulfuric acid, which would have leached the calcium from the phosphates [33]. Shells of fossil brachiopods are usually preserved as casts and molds, and when a rock containing the fossil brachiopod is cracked open, the split is usually along the layering plane that follows the internal or external shell surface. *Xianshanella haikouensis* exclusively occurs only as partially preserved body fossils, due to the distinctive and random cleavage across the valve and infilling sediment (Figure 1). The occurrence of this type of splitting is also possibly the result of a relatively lower degree of shell mineralization. The marginal setae of brachiopods are usually preserved with a distinct pseudomorphed framboidal texture in the Chengjiang Lagerstätte [33]. In our material of *X. haikouensis*, both branched fringes and marginal setae are preserved as aggregations of micro globular crystals without a distinct structure. This suggests that the fossil likely underwent extensive weathering during late diagenesis.

Chitinous setae, which are characteristic features observed in both annelids and brachiopods, along with Cambrian mickwitziids and tommotiids, have been proposed as a potential morphological synapomorphy among Cambrian lophotrochozoans [14]. Furthermore, annelids not only exhibit morphologically analogous ultrastructural setae to those of brachiopods, but they also commonly manifest branched (forked) setae, which, to date, have not been documented in other lophotrochozoans. Branched setae in *Orbinia latreilliid* and *Orbinia bioreti* (Annelida: Orbiniidae) were first reported by Hausam and Bartolomaeus [39], with further details provided by Tilic et al. [40]. These branched setae exhibit two distally separated tines that converge basally into a single setal shaft [39]. The formation of the branched setae has been described as a result of modulations of the microvilli pattern on the chaetoblast surface [39]. Despite notable differences in the cross-sectional architecture of individual setae between annelids and brachiopods, as well as variations in their branched morphologies, these structures remain comparable (see Figure 5). Recent molecular studies, including analyses of Hox genes (*lab* and *Post1*) and homeodomain gene (*Arx*) expression, further reveal conserved molecular signatures underpinning setal development in both annelids and brachiopods [14]. Thus, if the branched fringes of *X. haikouensis* are interpreted as elongated marginal setae, the convergence of morphological and molecular characteristics among branched and unbranched setae in these groups substantiates the hypothesis of setal homology between brachiopods and annelids.

The setae in *X. haikouensis* are coarser and thicker than those of other brachiopod species from the Chengjiang Lagerstätte [31]. This robust morphology likely enhances their protective function against durophagous predation while also reducing the risk of clogging their filter chamber with suspended particulate matter, while the comparatively slender setae of other species may primarily serve as tactile sensory extensions [31,41]. The branched fringes observed in *X. haikouensis* significantly extend the marginal setae, rendering their length comparable to that of the shell, and when these branches intersect, they create a dense barrier. Such intricate setal fringes are posited to have augmented the sensory capabilities of brachiopods while simultaneously offering enhanced protective measures for the organism. Furthermore, a multitude of rounded organic particles (approximately 200–500 μm in diameter) are dispersed randomly throughout the entire valve (Figure 1A and Figure 3). The presence of these particles, which are also preserved in the form of iron oxides, indicates that the elongated setal fringes may have served as fine sieves for the entrapment of organic particles, potentially aiding in filter feeding.

### 4.2. Branched Fringes as Attached Macroalgae and Their Ecological Implications

Examining exceptionally preserved Cambrian fossils yields extensive information and abundant fossil evidence for reconstructing both the life habits of individual taxa and the ecological dynamics of their communities [42,43,44,45,46,47,48,49,50]. Nevertheless, aspects of organismal ecology, such as group-level behaviors [51], still demand detailed ecological investigation. Insights gleaned from the Chengjiang fossil assemblage demonstrate that constituent organisms adopted diverse life modes, including infaunal, sessile epibenthic, vagile epibenthic, tubicolous, nektonic, planktonic, and pseudoplanktonic strategies [42,43,44,45,46,47,48,49,50,51,52,53]. Analyses of these ecological categories indicate that, in the wake of the Cambrian explosion, virtually all major trophic structures and life habits observable in modern biotas originated and underwent rapid diversification [42,53,54].

There are many cases in the Cambrian of organisms attaching onto other organisms [55,56,57,58,59,60,61,62,63,64], with examples including small tubular organisms attaching to Cambrian enigmatic animal *Vetulicola* [55], tubicolous cnidarian attaching to tubular fossil *Byronia* [56], unknown epibionts attaching to echinoderms [57]. Brachiopods were among the earliest benthic metazoans to establish ecological tiering, not only including primary tiers directly interacting with the seafloor but also explosively developing secondary and medium-high levels of tiering that relied on other biological debris [7]. They attached their shells to sponges and used elongated setae to mimic sponge spicules, camouflaging themselves to evade predators [7]. Fossil evidence from the Chengjiang Lagerstätte shows that the minute brachiopod *Kuangshanotreta malungensis* attached its shell to large macroalgae or graptolite branches, an association that resembled apples hanging from a tree [26]. Moreover, the brachiopod *Acrothele granulata* provided a substrate for sessile graptolites to attach to, with fossil evidence from the middle Cambrian in southern Sweden [65].

Since the delicate branched setae observed in this study have no prior reported occurrences in other brachiopods, we alternatively propose that these structures may represent attached macroalgae preserved at the distal ends of the marginal setae (Figure 6). However, macroalgae exhibiting this morphology have not been documented in the Chengjiang Lagerstätte to date [34,66]. Interestingly, the Ediacaran Miaohe biota has yielded macrofossils that show a similar morphology of the branched fringes [67,68]. *Doushantuophyton lineare* Steiner, 1994 [69] is characterized by its rather thin (typically 0.04–0.2 mm wide), mostly straight or rigid, regular and dichotomous or pseudomonopodial branches, though *D*. *lineare* has not been found in Chengjiang and other Cambrian Lagerstätten in South China. If correctly interpreted, these findings would represent a significant example supporting the complexity of early Cambrian benthic ecosystems. They provide evidence of secondary tiering, an ecological strategy dependent on biological debris, while demonstrating that brachiopods not only attached to other organisms but also served as substrates for epibionts, potentially facilitating mimicry as an anti-predatory adaptation. However, additional specimens of *X. haikouensis*, particularly those preserving delicate setal fringes, are critically needed to validate this hypothesis.

## 5. Conclusions

The discovery of enigmatic branched fringes in the early Cambrian lingulid brachiopod *Xianshanella haikouensis* from the Chengjiang Lagerstätte provides new insights into the morphological diversity and ecological complexity of early Cambrian brachiopods. These structures, preserved as iron oxides, exhibit intricate branched patterns and a length comparable to the shell, raising questions about their identity and function. Two primary hypotheses are proposed: (1) the branched fringes represent elongated setae, homologous to those found in annelids, supported by morphological similarities or (2) the structures are remnants of attached macroalgae, suggesting a mimicry strategy or ecological interaction previously undocumented in Chengjiang brachiopods.

If confirmed as setae, the branched fringes would underscore the homology of brachiopod and annelid setae, reinforcing their shared evolutionary history within the lophotrochozoan clade. Alternatively, if the structures are algal attachments, they would highlight the ecological versatility of brachiopods as substrates for other organisms, adding to the evidence of complex benthic interactions during the Cambrian Explosion. Further research, including advanced imaging techniques like synchrotron X-ray tomography and the discovery of additional specimens, is essential to resolve this puzzle. Regardless of their identity, these structures exemplify the exceptional preservation of the Chengjiang Biota and its role in illuminating the early evolution of animal morphology and ecology. This study not only expands our understanding of early Cambrian brachiopods but also emphasizes the importance of interdisciplinary approaches in paleontological research to unravel the mysteries of ancient life.

## Figures and Tables

**Figure 1 biology-14-01287-f001:**
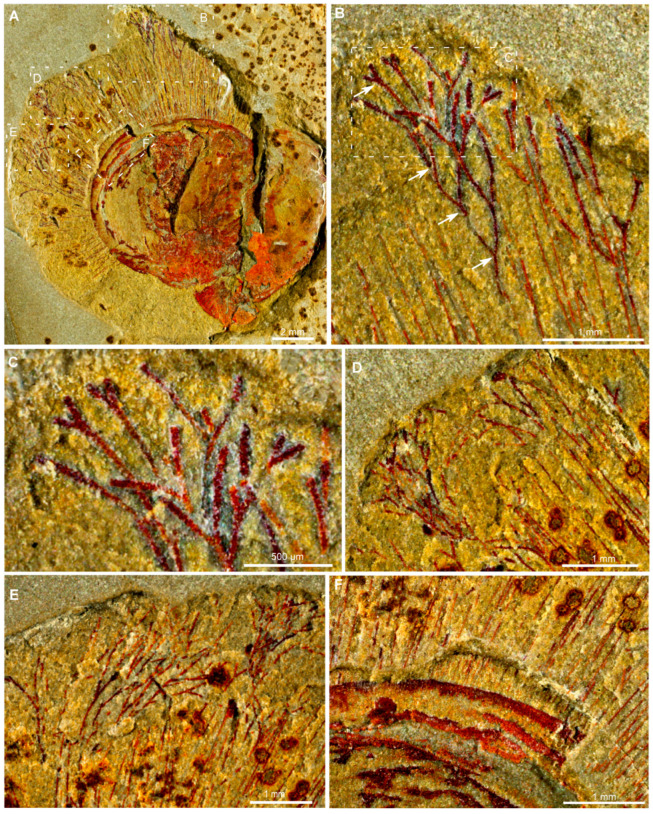
Plan view of *Xianshanella haikouensis* with branched fringes and marginal setae from the Chengjiang Lagerstätte, ELI 1781. (**A**–**F**) Details of branched fringes and marginal setae; white arrow in (**B**) indicates the base of the bifurcation.

**Figure 2 biology-14-01287-f002:**
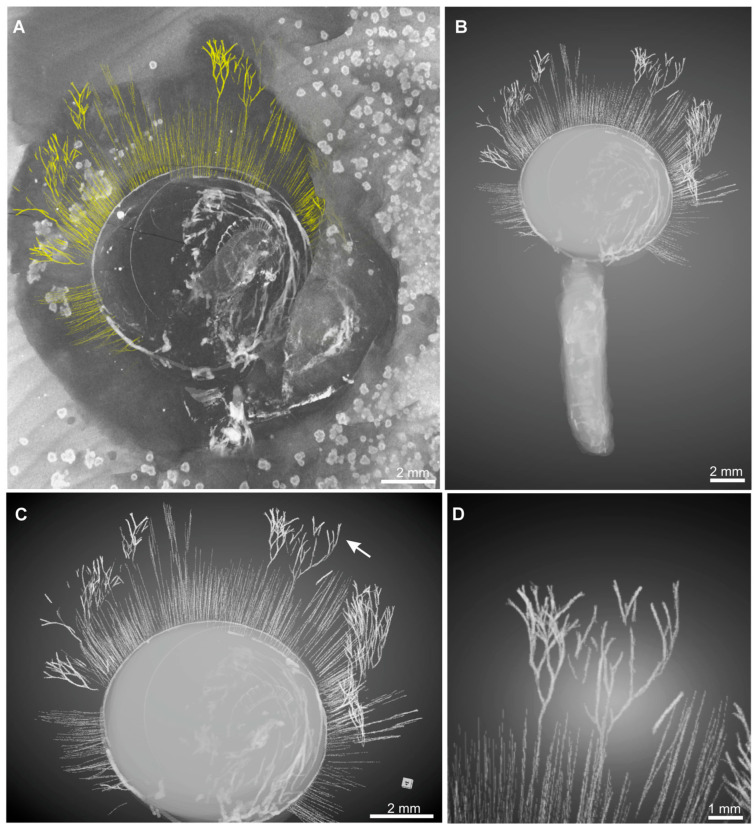
Three-dimensional Micro-CT reconstruction of the early Cambrian lingulid brachiopod *Xianshanella haikouensis*, illustrating its branched fringes and marginal setae, ELI 1781. (**A**) Overall view of the specimen and the surrounding matrix. The delicate structures of the marginal setae and branched fringes are digitally highlighted in yellow for clarity. (**B**) Reconstruction of the brachiopod, primarily showcasing its elongate and stout pedicle extending from the ventral valve. (**C**,**D**) Detailed views of the valve highlighting the long and branched fringes. The white arrow in (**C**) indicates the branching pattern at the anterior margin.

**Figure 3 biology-14-01287-f003:**
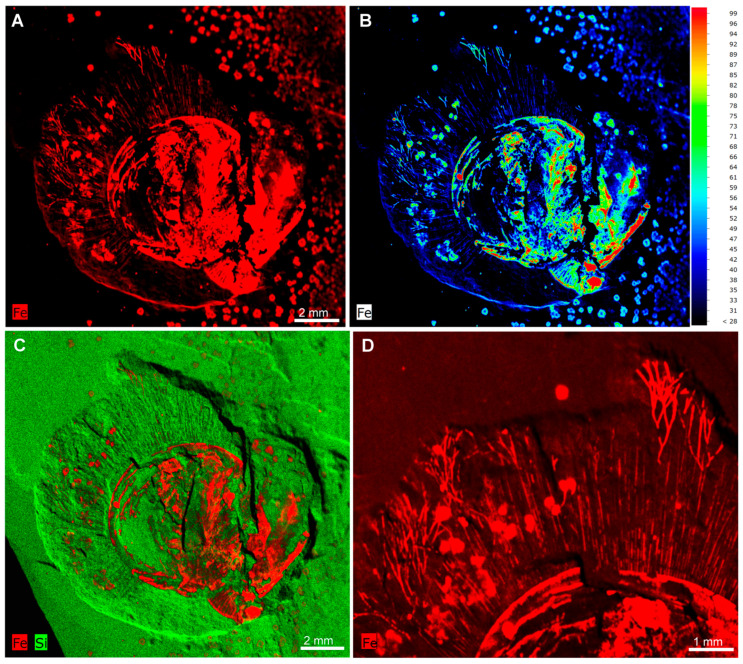
Micro XRF maps of the early Cambrian lingulid brachiopod *Xianshanella haikouensis* with branched fringe and marginal setae, ELI 1781. (**A**–**D**) Elemental abundances of iron (Fe) and silicon (Si) in the specimen. An enlarged view in (**D**) shows the highlighted branched fringes.

**Figure 4 biology-14-01287-f004:**
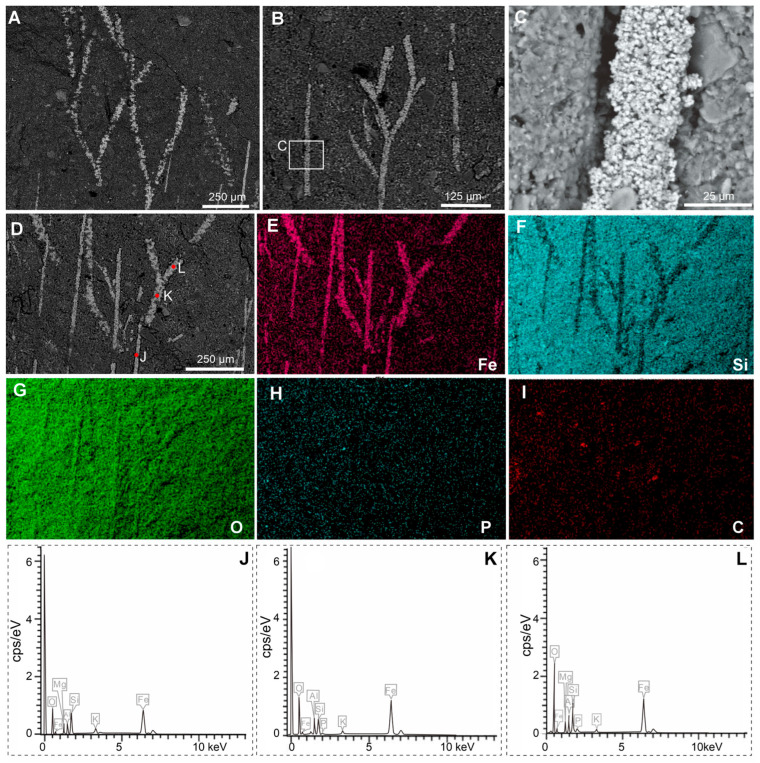
SEM and EDS results of the early Cambrian lingulid brachiopod *Xianshanella haikouensis* with branched fringes and marginal setae, ELI 1781. Both branched fringes and marginal setae are preserved as iron oxides, composed of aggregated iron oxide framboids. (**A**–**D**) BSE images of the branched fringes. (**E**–**I**) EDS elemental maps of the area in (**D**), showing that the marginal setae and branched fringes are deficient in silicon (**F**) but enriched in iron (**E**) and oxygen (**G**), with weak signals for phosphorus (**H**) and carbon (**I**). (**J**–**L**) Energy dispersive X-ray spectrograph of points indicated in (**D**) highlight compositional differences in the marginal seta (**J**) and branched fringes (**K**,**L**).

**Figure 5 biology-14-01287-f005:**
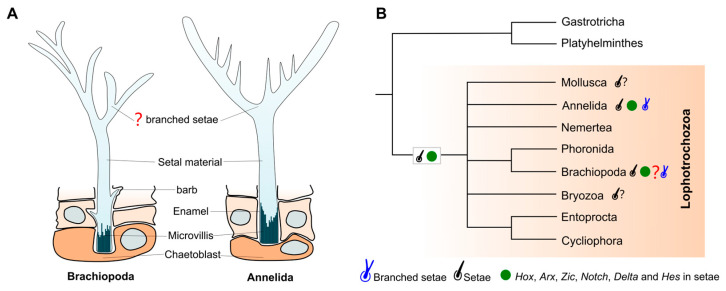
Morphological similarities and phylogenetic relationships of setae between brachiopods and annelids. (**A**) Comparative morphology and secretion of branched setae in brachiopods and annelids. Drawing adapted with permission from ref. [23]. (**B**) Phylogenetic distribution of setal characters showing shared morphological and molecular features in Brachiopoda and Annelida (red “?” indicates potential branched setae in Brachiopoda; black “?” denotes setae-like structures in Mollusca and Bryozoa), supporting the homology of this lophotrochozoan trait.

**Figure 6 biology-14-01287-f006:**
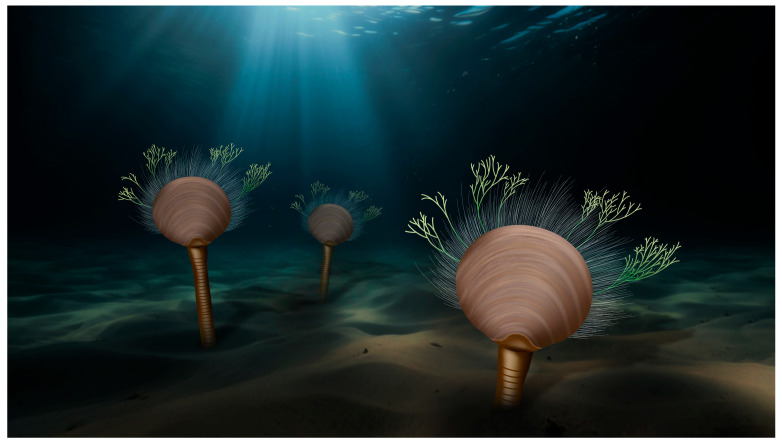
An artistic reconstruction depicting macroalgae attached to the lingulid brachiopod *Xianshanella haikouensis*.

## Data Availability

Specimen is deposited in the State Key Laboratory of Continental Evolution and Early Life, Shaanxi Key Laboratory of Early Life and Environments, Department of Geology, Northwest University, Xi’an 710069, China.

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
