# Peer review of "Branched Setae or Attached Macroalgae: A Case Study of an Exceptionally Preserved Brachiopod from the Early Cambrian Chengjiang Lagerstätte"

_biology, 2025, doi:10.3390/biology14091287_

Round 1

Reviewer 1 Report

Comments and Suggestions for Authors The study presents a thoughtful and well-executed investigation into branched setae in brachiopods, which holds significant anatomical and ecological implications. While fossil records of brachiopod setae are exceptionally rare, this work rigorously evaluates the potential presence of branched structures based on a single, exceptionally preserved specimen, supported by multiple analytical techniques. The authors cautiously propose the hypothesis of branched setae while explicitly acknowledging the alternative interpretation that the structures may represent macroscopic algal remains. This research offers a meaningful contribution to our understanding of brachiopod morphology and paleoecology. The following revisions are recommended to strengthen the manuscript.
1) Given that the entire analysis is based on a single specimen, it is essential to establish a clear and direct visual linkage between Figure 1 and Figure 2 (and Figure 3 if applicable). Although the region corresponding to Figure 1B is visible in Figure 2B, the correspondence is not immediately apparent. It is recommended that the same area in Figure 2B be explicitly outlined (e.g., with a box or annotation) and clearly labeled as equivalent to Figure 1B. This would help readers confirm that the morphological features observed in Figure 1 are indeed the same structures visible in the Micro-CT reconstruction in Figure 2, especially since the pedicle region is completely absent in Figure 1.
2) Typical branched setae usually taper (or at least maintain constant diameter) toward the distal end, yet the observed structures appear to thicken. This morphological discrepancy warrants further discussion. Moreover, the Micro-CT images show differences in clarity between the setae and the branched parts. This may be due to diameter variation, but could also reflect compositional differences. Considering that the current Micro-XRF data consist of only one image, it is recommended to include additional localized high-resolution images of the branched regions, if available, to better support elemental consistency.
3) The hypothesis that elongated setal fringes may have functioned as fine sieves for organic particles (line 224) is a reasonable and interesting observation, and is supported by the anatomical arrangement in Figure 2 (as brachiopod water flow is in from both sides and out from the front center part). However, the alternative interpretation involving macroalgae is less convincing, as the attachment site appears to be spatially selective and lacks clear ecological context. If possible, a brief discussion would help further assess the plausibility of this alternative hypothesis.   4) The authors mention that the fossil "likely underwent extensive weathering" (line 187) and the presence of iron oxides, which raises the possibility that the branched structures may represent pseudofossils. These typically form as 2D surface features, not as three-dimensional substructures. To rule this out, a transverse Micro-CT section through the branched region is strongly recommended.

Author Response

Comments 1: The study presents a thoughtful and well-executed investigation into branched setae in brachiopods, which holds significant anatomical and ecological implications. While fossil records of brachiopod setae are exceptionally rare, this work rigorously evaluates the potential presence of branched structures based on a single, exceptionally preserved specimen, supported by multiple analytical techniques. The authors cautiously propose the hypothesis of branched setae while explicitly acknowledging the alternative interpretation that the structures may represent macroscopic algal remains. This research offers a meaningful contribution to our understanding of brachiopod morphology and paleoecology. The following revisions are recommended to strengthen the manuscript.

Response 1: We are deeply grateful to the reviewer for their valuable time and the thoughtful review of our work. Their positive feedback and constructive suggestions are immensely appreciated.

Comments 2: Given that the entire analysis is based on a single specimen, it is essential to establish a clear and direct visual linkage between Figure 1 and Figure 2 (and Figure 3 if applicable). Although the region corresponding to Figure 1B is visible in Figure 2B, the correspondence is not immediately apparent. It is recommended that the same area in Figure 2B be explicitly outlined (e.g., with a box or annotation) and clearly labeled as equivalent to Figure 1B. This would help readers confirm that the morphological features observed in Figure 1 are indeed the same structures visible in the Micro-CT reconstruction in Figure 2, especially since the pedicle region is completely absent in Figure 1.

Response 2: We thank the reviewer for their careful reading and insightful suggestion. The reviewer is correct that our entire analysis is based on a single specimen. We have added information to the figure captions to indicate that all figures and results presented in this manuscript are based on a single specimen (specimen no. ELI 1781). We have also added more Micro-CT results in Figure 2, which exhibit an overall view of the brachiopod and its surrounding matrix, as well as an enlargement of the branched fringes at the anterior part.

Comments 3: Typical branched setae usually taper (or at least maintain constant diameter) toward the distal end, yet the observed structures appear to thicken. This morphological discrepancy warrants further discussion. Moreover, the Micro-CT images show differences in clarity between the setae and the branched parts. This may be due to diameter variation, but could also reflect compositional differences. Considering that the current Micro-XRF data consist of only one image, it is recommended to include additional localized high-resolution images of the branched regions, if available, to better support elemental consistency.

Response 3: We thank the reviewer for their thoughtful suggestion. In response, we have added more Micro-XRF data and created a new Figure 3 composed exclusively of these results. The new Figure 3D highlights the branched region. The results indicate that both the branched and unbranched regions are preserved as iron oxides, similar to the brachiopod valve. However, the branched fringes show a higher concentration of iron compared to the marginal setae, which likely provide more evidence for the macroalgae attachment hypothesis.

Comments 4: The hypothesis that elongated setal fringes may have functioned as fine sieves for organic particles (line 224) is a reasonable and interesting observation, and is supported by the anatomical arrangement in Figure 2 (as brachiopod water flow is in from both sides and out from the front center part). However, the alternative interpretation involving macroalgae is less convincing, as the attachment site appears to be spatially selective and lacks clear ecological context. If possible, a brief discussion would help further assess the plausibility of this alternative hypothesis. 

Response 4: We acknowledge the reviewer’s professionalism and careful review. It is true that, based on the authors’ previous research, water flow was inhaled from both sides and exhaled from the front central part, and the branched fringe may represent a component of the setae. However, this hypothesis requires more evidence for confirmation. Branched setae in brachiopods have never been reported from any time period or fossil deposit, and the wider diameter of the branched fringe makes this hypothesis more obscure. Furthermore, numerous studies have reinforced the ecological complexity of brachiopods from Cambrian Lagerstätten, such as the Chengjiang and Guanshan biotas from South China and the Burgess Shale from Canada. Our results are highly comparable to the mimicry reported from the Burgess Shale by our co-author, Timothy Topper (Ref. 11). Accordingly, we present both hypotheses but provide stronger support for the hypothesis of macroalgae attachment. We also emphasize that more fossil samples and advanced techniques are needed to resolve this question.

Comments 5: The authors mention that the fossil "likely underwent extensive weathering" (line 187) and the presence of iron oxides, which raises the possibility that the branched structures may represent pseudofossils. These typically form as 2D surface features, not as three-dimensional substructures. To rule this out, a transverse Micro-CT section through the branched region is strongly recommended.

Response 5: We thank the reviewer for this thoughtful suggestion. It is true that the fossil likely underwent extensive weathering, as is the case for most of fossil material from exceptional preservation deposits (Lagerstätten) such as the Chengjiang and Guanshan biotas. However, the branched fringe is unlikely to be a pseudofossil. In contrast, it exhibits a microstructure similar to both the marginal setae of this specimen (Fig. 4A-E) and other Chengjiang brachiopods. For a detailed comparison, please refer to our most recent research article on the setae of brachiopods and other lophotrochozoans (Ref. 17). We also attempted to locate a transverse Micro-CT section through the branched region; unfortunately, the results were not promising. Although the marginal setae and branched fringe were lightly mineralized and preserved as iron oxides during diagenesis, they form very thin films that are below detection limits. As an alternative, we have added more Micro-CT results (Fig. 2A, D) to better highlight the morphology of the marginal setae and branched fringes.

4. Response to Comments on the Quality of Figures

Point 1: Figures and tables must be improved

Response 1: We have added more Micro-CT results to Figure 2 and created a new figure for the Micro-XRF data (Figure 3).

Reviewer 2 Report

Comments and Suggestions for Authors

The article is an excellent contribution to understanding new paleoecological patterns of the Cambrian biota. The paper is very interesting and well-written. Besides, the methods and pictures are really good. I recommend its publication, but some minor corrections are needed before that. 

1) In the Materials and Methods section, the word specimens is very repetitive. It could be sample or material

2) Improve graphics in Figure 3, since the elements and words in general are very tiny; moreover, the graphs are slightly blurred. 

What is the main question addressed by the research?
The main question is to understand the structures of the spines, specifically whether they are setae or some algae development. This is important because the setae would indicate a closer relation between brachiopods and annelids, which are part of the clade Lophotrochozoa. Besides, this would complement more information about the clade.

Do you consider the topic original or relevant to the field?
I consider the contribution interesting and significant. Given the preservation of the fossil studied, even if a Cambrian sample, it allows the study of the enigmatic structures. The relevance lies in the fact that the specimen studied is the only one with these structures preserved. Moreover, the work contributes to the knowledge of the evolution of organisms with setae.

Does it address a specific gap in the field?
Yes, because the evolution and links among animals are an essential topic to understand Earth's life. Thus, the possible presence of setae on a brachiopod could prove the relation of these animals with annelids. Given that both groups belong to Lophotrochozoa, the setae character may be a synapomorphy. 

What does it add to the subject area compared with other published material?
The preservation of the sample is exquisite, since both valves with their spines, as well as the pedicle, are well preserved; besides, the tomography displayed the form of the lophophore and mantle channels. Even the possible setae or algae could be studied in detail. Such preservation allowed the detailed study of the structures, discarding different possibilities by means of various analyses.

What specific improvements should the authors consider regarding the methodology?
None, I think that the authors defended their proposals well.

Are the conclusions consistent with the evidence and arguments presented and do they address the main question posed?

The conclusions were consistent with the methodology and evidence, since both proposals (setae or algae) were proven by means of distinct studies. Even the contribution includes why the structures could be setae or algae. I believe this work provides a solid foundation for understanding the similarities among lophotrochozoans more effectively.

Are the references appropriate?
Yes, these are recent and support the study.

Any additional comments on the tables and figures.
As mentioned first, figures 3J, 3K, and 3L must be improved, given the letters are tiny. Moreover, figures are blurry.

Author Response

Comments 1: The article is an excellent contribution to understanding new paleoecological patterns of the Cambrian biota. The paper is very interesting and well-written. Besides, the methods and pictures are really good. I recommend its publication, but some minor corrections are needed before that.

Response 1: The authors extend their sincere gratitude to the reviewer for the time and effort in reviewing our manuscript. We are also greatly encouraged by their positive feedback, which has been invaluable in strengthening this work.

Comments 2: In the Materials and Methods section, the word specimens are very repetitive. It could be sample or material

Response2: Corrected.

Comments 3: Improve graphics in Figure 3, since the elements and words in general are very tiny; moreover, the graphs are slightly blurred.

Response 3: Corrected. We have re-arranged and redrawn Fig. 3 J, K, and L. It looks much clearer now.

Comments 4: What is the main question addressed by the research?

The main question is to understand the structures of the spines, specifically whether they are setae or some algae development. This is important because the setae would indicate a closer relation between brachiopods and annelids, which are part of the clade Lophotrochozoa. Besides, this would complement more information about the clade.

Response 4: Thanks!

Comments 5: Do you consider the topic original or relevant to the field?

I consider the contribution interesting and significant. Given the preservation of the fossil studied, even if a Cambrian sample, it allows the study of the enigmatic structures. The relevance lies in the fact that the specimen studied is the only one with these structures preserved. Moreover, the work contributes to the knowledge of the evolution of organisms with setae.

Response 5: Thanks!

Comments 6: Does it address a specific gap in the field?

Yes, because the evolution and links among animals are an essential topic to understand Earth's life. Thus, the possible presence of setae on a brachiopod could prove the relation of these animals with annelids. Given that both groups belong to Lophotrochozoa, the setae character may be a synapomorphy.

Response 6: Thanks!

Comments 7: What does it add to the subject area compared with other published material?

The preservation of the sample is exquisite, since both valves with their spines, as well as the pedicle, are well preserved; besides, the tomography displayed the form of the lophophore and mantle channels. Even the possible setae or algae could be studied in detail. Such preservation allowed the detailed study of the structures, discarding different possibilities by means of various analyses.

Response 75: Thanks!

Comments 8: What specific improvements should the authors consider regarding the methodology?

None, I think that the authors defended their proposals well.

Response 8: Thanks!

Comments 9: Are the conclusions consistent with the evidence and arguments presented and do they address the main question posed?

The conclusions were consistent with the methodology and evidence, since both proposals (setae or algae) were proven by means of distinct studies. Even the contribution includes why the structures could be setae or algae. I believe this work provides a solid foundation for understanding the similarities among lophotrochozoans more effectively.

Response 9: Thanks!

Comments 10: Are the references appropriate?

Yes, these are recent and support the study.

Response 10: Thanks!

Comments 11: Any additional comments on the tables and figures.

As mentioned first, figures 3J, 3K, and 3L must be improved, given the letters are tiny. Moreover, figures are blurry.

Response 11: We thank the reviewer for all of the positive feedback. Figure 3 has been revised according to the reviewer’s suggestions. It looks much clearer now.

Reviewer 3 Report

Comments and Suggestions for Authors

Title

Consider changing the title. Maybe like this one: 

Attached macroalgae or branched setae. Resolving the identity of structures in early Cambrian brachiopod.

Simple Summary

Rewrite The SS section. It should be shorter. The language should be plain. Phrases like “Shedding light on how early life forms were diversified@ and “ showing how organisms interacted and relied one on another for survival” are speculative and overstated due to the evidence is based on single specimen.

Lines 23-28 These sentences are long and complex, break them into several simpler sentences.

Please, present your results first, and then state scientific debates. Now the text concentrates on alternative hypotheses, which may confuse general readers. 

Abstract

The abstract is too long and dense with details. 

There is no clear IMRaD flow. The text jumps from background to detailed results without a smooth transition. The hypotheses on branched setae vs macroalgae are mixed into results and discussion without separation. The macroalgae hypothesis that  was dismissed.

The main takeaway is missed in too many details. 

Introduction

The section is too long and overloaded with citations. Detailed discussion of embryonic and larval features, hox gene expression appears too early. A reader may lose focus.

The research gap is unclear. 

The Introduction does not highlight why branched fringes of Xianshanella haikouensis are novel until late. 

The section jumps between broad context and narrow details without smooth transition.

Materials and methods

This section should be restructured.

The section reads like a list of tools and instruments, rather than structured methodology. 

Write about sample preparation, scanning parameters, thresholds, technical procedures.

Why was each method chosen? It’s unclear why to use microCT scanning while SEM revealed preservation. 

Please, provide accession| catalogue number of fossils. Did you examine several specimens or only one? Clarify that moment.

Provide details of scanning resolution, voxel size, energy settings coating for SEM and reconstruction parameters for Micro-CT.

The methods presented as a chronological list, but not grouped by type ( microscopy, imaging, tomography, geochemical analysis). It’s difficult to follow. 

Please, provide details for stratigraphic context - what exact stratigraphic unit? Any correlations with previous reports?

Results

The section reads like a list of measurements: valve length|width, setae diameters, branching angles, bifurcation heights. This overwhelms the reader. Move these data to the table, or plot< or supplementary. 

The data presented as raw descriptions without biological patterns. Physiological or ecological relevance of size difference is not discussed there.

All comparisons are qualitative and lack statistical support. Please, correct it. Add range, standard deviations, sample size.

The results section does not emphasize that research was done on a single individual. 

Figures are not fully described. The explanation of elemental maps (fig 3) is very brief.

Some phrases “making it challenging to distinguish between dorsal and ventral valve setal arrangements “ are more relevant for discussion.

Figures 

In some legend Fig 1 the descriptions are scarce and don't explain what the reader should notice.

Figures 1-5 are described as containing multiple panels, however the text does not explain whether labels, scale bars, and contrast are clear. Micrographs often require high resolution.

Fig 3 combines too many elements, BSE images and multiple elemental maps. This may overwhelm the reader. 

fig 1-3 add orientation aids. It’s unclear whether arrows, labels  and color overlays are consistently used.

Elemental data presented visually, but not quantitative values or spectra are shown. 

Discussion

Rewrite the discussion. It’s too long and fragmented, repetitive and speculative. Both hypotheses are interesting, however they need clear framing. Some claims go further rather than evidence support. Suggestion that brachiopod and annelid setae are homologous is strong given that the evidence is from a single fossil.Macroalgae hypothesis is also speculative, without direct fossil evidence of macroalgae in Chengjiang.

Conclusions

Rewrite this section. Now it repeats results and discussion. 

Sentences are long and technical. Please, simplify them.

Reference list

Please, double check formatting. Some journals contain Full titles, while others use abbreviations. 

Many citations are from the author's own group. Numbers 3,4,11,12,17,30 and 33. It’s relevant, however, there is a risk of self-citation bias. Comparative annelid setae studies is limited by a couple of sources, please, add modern research. It will strengthen the manuscript.

Some citations in Introduction and discussion 15-25 are clustered, however not clearly correspond to the arguments. 

Author Response

Comments 1: Title. Consider changing the title. Maybe like this one: Attached macroalgae or branched setae. Resolving the identity of structures in early Cambrian brachiopod.

Response 1: We thank the reviewer for this suggestion. We have considered it carefully but feel our original title is preferable because it underscores the importance of this example of exceptional preservation from the Chengjiang Lagerstätte, a World Heritage Site that is critical to advancing our understanding of early life evolution.

Comments 2: Simple Summary. Rewrite The SS section. It should be shorter. The language should be plain. Phrases like “Shedding light on how early life forms were diversified” and “showing how organisms interacted and relied one on another for survival” are speculative and overstated due to the evidence is based on single specimen. Lines 23-28 These sentences are long and complex, break them into several simpler sentences. Please, present your results first, and then state scientific debates. Now the text concentrates on alternative hypotheses, which may confuse general readers. 

Response 2: We have modified the content in the Simple Summary section to improve its clarity and readability. The changes are highlighted in blue in the main text.

Comments 3: Abstract. The abstract is too long and dense with details. There is no clear IMRaD flow. The text jumps from background to detailed results without a smooth transition. The hypotheses on branched setae vs macroalgae are mixed into results and discussion without separation. The macroalgae hypothesis that was dismissed. The main takeaway is missed in too many details. 

Response 3: We have revised the Abstract section. We proposed two hypotheses about these branched fringes and provided arguments from multiple perspectives. We are more inclined to support the macroalgae hypothesis and have provided more arguments for it.

Comments 4: Introduction. The section is too long and overloaded with citations. Detailed discussion of embryonic and larval features, hox gene expression appears too early. A reader may lose focus. The research gap is unclear. The Introduction does not highlight why branched fringes of Xianshanella haikouensis are novel until late. The section jumps between broad context and narrow details without smooth transition.

Response 4: We thank the reviewer for this helpful suggestion. We have revised the introduction section, and it is more readable now. 

Comments 5: Materials and methods. This section should be restructured. The section reads like a list of tools and instruments, rather than structured methodology. Write about sample preparation, scanning parameters, thresholds, technical procedures. Why was each method chosen? It’s unclear why to use microCT scanning while SEM revealed preservation. Please, provide accession| catalogue number of fossils. Did you examine several specimens or only one? Clarify that moment. Provide details of scanning resolution, voxel size, energy settings coating for SEM and reconstruction parameters for Micro-CT. The methods presented as a chronological list, but not grouped by type (microscopy, imaging, tomography, geochemical analysis). It’s difficult to follow. Please, provide details for stratigraphic context - what exact stratigraphic unit? Any correlations with previous reports?

Response 5: We have thoroughly revised the Materials and Methods section. The specimen of Xianshanella haikouensis was recovered from the Chengjiang Lagerstätte. Detailed stratigraphic and section information is derived from Zhang et al. [13]. To maintain conciseness, we have not repeated these details in the manuscript. Regarding methodology, we have added the scanning parameters for SEM analyses. We also included the reasons for using Micro-CT, SEM, and Micro-XRF. Furthermore, we added more Micro-CT results to Figure 2 and created a new figure for the Micro-XRF data (Figure 3).

Comments 6: Results. The section reads like a list of measurements: valve length|width, setae diameters, branching angles, bifurcation heights. This overwhelms the reader. Move these data to the table, or plot< or supplementary. The data presented as raw descriptions without biological patterns. Physiological or ecological relevance of size difference is not discussed there. All comparisons are qualitative and lack statistical support. Please, correct it. Add range, standard deviations, sample size. The results section does not emphasize that research was done on a single individual. Figures are not fully described. The explanation of elemental maps (fig 3) is very brief. Some phrases “making it challenging to distinguish between dorsal and ventral valve setal arrangements “ are more relevant for discussion.

Response 6: We apologize for the lack of clarity in this section. Our description begins with the morphology of the specimen, including the size of shell, marginal setae and branched fringes, following standard paleontological practices. We then present our Micro-CT results, which reveal previously obscured details of the pedicle and lophophore that were concealed by the surrounding matrix. Subsequently, to determine whether the branched and unbranched regions share a similar chemical composition, we present our Micro-XRF and EDS results. We added specimen number in Figures 1-4 to indicate that research was done on a single individual. And we also added more information on figure caption for Fig. 4 (previous Fig 3).

Comments 7: In some legend Fig 1 the descriptions are scarce and don't explain what the reader should notice. Figures 1-5 are described as containing multiple panels; however the text does not explain whether labels, scale bars, and contrast are clear. Micrographs often require high resolution.

Response 7: We added more details on all figure captions, emphasizing that all results are recovered from one specimen of Xianshanella haikouensis, specimen number ELI 1781. All figures are provided with high quality and meet the journal’s requirements about image resolution.

Comments 8: Fig 3 combines too many elements, BSE images and multiple elemental maps. This may overwhelm the reader. 

Response 8: We thank the reviewer for this helpful comment, and we would like to keep these EDS elemental maps as they are important to show the elemental composition of the fossils and matrix. We have redrawn the EDS spectrum and it is much clearer now.

Comments 9: fig 1-3 add orientation aids. It’s unclear whether arrows, labels and color overlays are consistently used. Elemental data presented visually, but not quantitative values or spectra are shown. 

Response 9: The orientation of the fossil in Figure 1 is now explicitly indicated by the term "plan view" in its caption. Furthermore, the EDS spectrum in Figure 4 has been redrawn to enhance its legibility.

Comments 10: Discussion. Rewrite the discussion. It’s too long and fragmented, repetitive and speculative. Both hypotheses are interesting, however they need clear framing. Some claims go further rather than evidence support. Suggestion that brachiopod and annelid setae are homologous is strong given that the evidence is from a single fossil. Macroalgae hypothesis is also speculative, without direct fossil evidence of macroalgae in Chengjiang.

Response 10: We have revised this section. Based on our multi-disciplinary analysis, we first evaluated the hypothesis that the branched fringes represent elongated setae and discussed its phylogenetic implications. However, the significant difference in diameter between the branched fringes and the marginal setae makes it difficult to support this idea. We then discuss the alternative hypothesis that the structures are attached macroalgae. We find this hypothesis more compelling, not only due to the diametrical differences mentioned above but also because such enigmatic branched setae have never been reported in any brachiopods. Finally, we emphasize that more fossil material is needed to resolve this paradox conclusively.

Comments 11: Conclusions. Rewrite this section. Now it repeats results and discussion. Sentences are long and technical. Please, simplify them.

Response 11: This section is the summarized conclusions based on our results and discussion. We have modified this section and it is more readable now.

Comments 12: Please, double check formatting. Some journals contain Full titles, while others use abbreviations. Many citations are from the author's own group. Numbers 3,4,11,12,17,30 and 33. It’s relevant, however, there is a risk of self-citation bias. Comparative annelid setae studies is limited by a couple of sources, please, add modern research. It will strengthen the manuscript. Some citations in Introduction and discussion 15-25 are clustered, however not clearly correspond to the arguments. 

Response 12: We thank the reviewer for this careful suggestion. We have revised the reference list accordingly. Less relevant self-citations from our co-authors were removed prior to the current revision, as requested by the editor. The current self-citation rate is now well below 15%, which meets the journal’s requirements. We have also double-checked the reference formatting for consistency. Furthermore, we have ensured that the most recent comparative studies on annelid setae are included, such as Liang et al., 2025 [17] and Schiemann et al., 2017 [18]. References 15-25 are strategically cited throughout the Introduction and Discussion sections. Furthermore, we have added five new references (61-64, 68) that provide evidence of attachment strategies during the Cambrian period, which will serve to support our findings. These literatures provide essential foundational support for our research and are critical to the interpretation of Cambrian brachiopod fossils and their ecological implications.

4. Response to Comments on the Quality of English Language

Point 1: The English could be improved to more clearly express the research.

Response 1: The English was thoroughly revised by the second author, Timothy P. Topper, a native English speaker.

5. Response to Comments on the Quality of Figures

Point 1: Figures and tables must be improved

Response 1: We have added more Micro-CT results to Figure 2 and created a new figure for the Micro-XRF data (Figure 3).

Round 2

Reviewer 3 Report

Comments and Suggestions for Authors

I read the manuscript carefully. The authors did a great job. The text is written in clear language, grammar and flow were improved.

The Methods and Results sections are drastically improved in organisation. 

Figures and captures are more professional. Captions expanded and include context. 

Artistic reconstruction became clearer in better wording. Measurements and units are more precise. 

The discussion section was expanded, including more references. The authors engaged with the feedback. 

I think the paper is close to acceptance.